# Chlorodifluoromethane Hydrodechlorination on Carbon-Supported Pd-Pt Catalysts. Beneficial Effect of Catalyst Oxidation

Monika Radlik [1,*], Wojciech Juszczyk [2], Wioletta Raróg-Pilecka [3], Magdalena Zybert [3] and Zbigniew Karpiński [1,*]

1   Faculty of Mathematics and Natural Sciences, Cardinal Stefan Wyszyński University in Warsaw, ul. Wóycickiego 1/3, PL-01938 Warszawa, Poland

2   Institute of Physical Chemistry, Polish Academy of Sciences, ul. Kasprzaka 44/52, PL-01224 Warszawa, Poland; wjuszczyk@ichf.edu.pl

3   Faculty of Chemistry, Warsaw University of Technology, ul. Noakowskiego 3, PL-00664 Warszawa, Poland; wiola@ch.pw.edu.pl (W.R.-P.); mzybert@ch.pw.edu.pl (M.Z.)

*   Correspondence: m.radlik@uksw.edu.pl (M.R.); z.karpinski@uksw.edu.pl (Z.K.); Tel.: +48-516138358 (M.R.); +48-604053267 (Z.K.)

**Abstract:** Previously tested 2 wt % palladium-platinum catalysts supported on Norit activated carbon preheated to 1600 °C have been reinvestigated in $CHFCl_2$ hydrodechlorination. An additionally adopted catalyst oxidation at 350–400 °C produced nearly an order of magnitude increase in the turnover frequency of Pd/C, from $4.1 \times 10^{-4}$ to $2.63 \times 10^{-3}$ $s^{-1}$. This increase is not caused by changes in metal dispersion or possible decontamination of the Pd surface from superficial carbon, but rather by unlocking the active surface, originally inaccessible in metal particles tightly packed in the pores of carbon. Burning carbon from the pore walls attached to the metal changes the pore structure, providing easier access for the reactants to the entire palladium surface. Calcination of Pt/C and Pd-Pt/C catalysts results in much smaller evolution of catalytic activity than that observed for Pd/C. This shapes the relationship between turnover frequency (TOF) and alloy composition, which now does not confirm the Pd-Pt synergy invoked in the previous work. The absence of this synergy is confirmed by gradual regular changes in product selectivity, from 70 to 80% towards $CH_2F_2$ for Pd/C to almost 60% towards $CH_4$ for Pt/C. The use of even higher-preheated carbon (1800 °C), completely free of micropores, results in a Pd/C catalyst that does not need to be oxidized to achieve high activity and excellent selectivity to $CH_2F_2$ (>90%).

**Keywords:** $CHClF_2$ hydrodechlorination; Pd-Pt/C; thermally modified activated carbon; beneficial effect of catalyst oxidation; pore structure changes

## 1. Introduction

Despite the ban on the use of hydrochlorofluorocarbons (HCFCs) [1], HCFC-22 is still the most abundant HCFC in the atmosphere [2–5]. Huge amounts of chlorodifluoromethane originating from refrigerators and air conditioners must be destroyed or, preferably, transformed to other useful substances. Catalytic hydrodechlorination is a promising method for the beneficial utilization of detrimental Cl-containing compounds by converting them into valuable, non-hazardous chemicals [6–16]. This strategy was previously proposed for the elimination of dichlorodifluoromethane (CFC-12) by using supported metal catalysts, mainly palladium, where the formation of the environmentally friendly refrigerant HFC-32 ($CH_2F_2$), was achieved with great selectivities and yields, at relatively low reaction temperatures [6,7,9]. Palladium is also active in $CHClF_2$ hydrodechlorination, showing good product selectivity to the desired $CH_2F_2$ [6–8,14,15]. However, because of the lower reactivity of this reactant, significantly higher reaction temperatures should be used than in the case of dichlorodifluoromethane hydrodechlorination.

Recently, we published results on the hydrodechlorination of $CHClF_2$ over carbon-supported Pd-Pt alloy catalysts [17]. The purpose of selecting a highly preheated (1600 °C) commercial Norit carbon was to obtain a catalyst support with a high degree of purity and relatively free of micropores. Bimetallic Pd-Pt/C catalysts, prepared by impregnation, demonstrated better performance than the monometallic ones, which is in harmony with the previous results on hydrodechlorination of various chlorine-containing substances [9–12,15]. This synergistic effect was ascribed to charge transfer between Pt and Pd, in line with a generally adopted reaction mechanism of catalytic hydrodechlorination [18] and published data on the electronic structure of Pd-Pt alloys [19].

Although finding the synergy in $CHClF_2$ hydrodechlorination appeared encouraging [17], shortly thereafter we realized that the catalytic activity of carbon-supported palladium was found surprisingly low, at the level of the activity of platinum, regarded as the much less reactive metal in this reaction [15,16]. This raised concerns that the Pd/C catalyst could be partially deactivated. It should be stressed that, during our previous studies, we were well aware of the possibility of blocking the palladium surface by carbon from the support and removing this carbon from the palladium surface. According to suggestions resulting from earlier reports [20], and especially from more recent works [21,22], we pretreated our Pd-Pt/C catalysts with diluted oxygen at 300 °C for 1 or 2 h, prior to catalyst activation in $H_2$ [17]. It should also be noted that our highly preheated turbostratic carbon showed a considerable degree of graphitization [17], resulting in a less serious contamination of the deposited Pd by carbon from the support. As was established in [22], the degree of decoration of the Pd surface by carbon decreases with graphitization of the carbon supports due to stronger C–C interactions.

In this work, we decided to reinvestigate the Pd-Pt/C catalysts more thoroughly. This time, we decided to pretreat our samples at higher than 300 °C preoxidation temperatures, prior to catalyst reduction. This pretreatment will be referred to as the 'precalcination' along the text. Since the pretreatment resulted in a considerable improvement of catalytic performance (especially for Pd/C catalysts), we also performed different tests of catalyst characterization, evaluating for changes in metal dispersion, crystallite size of the metal phase, and changes in the pore structure produced by preoxidation. Temperature programmed oxidation (TPO) of carbon-supported metal catalysts was intended to provide information on the forms of carbon removed at different stages of catalyst precalcination [21].

## 2. Results and Discussion

Catalytic screening of both series (ex-acac and ex-chloride) of Pd-Pt/Norit1600 catalysts in chlorodifluoromethane hydrodechlorination showed that steady conversions, generally <3%, were attained after 16–20 h, as reported in our previous paper [17].

Detailed data showing the catalytic properties of Pd-Pt/Norit1600 catalysts precalcined at different temperatures are in Tables S1–S4, placed in the Supplementary Materials. The effect of precalcination temperature of the catalysts is demonstrated in Figures 1 and 2. As in our previous work [17], for the Pd-rich samples, $CH_4$ and $CH_2F_2$ (HFC-32) were found to be the predominant products, making up more than 90% of all products (Tables S1–S4 and Figure 1). For the monometallic platinum samples, $CHF_3$ and $CH_3F$ made up 30–40% of all products. During stabilization, variations in selectivity were rather small, although usually $CH_2F_2$ formation was increased at the expense of methane.

The product selectivity shown in Figure 1A–C indicates the best selectivity to $CH_2F_2$ for 2 wt %Pd*100*(acac)/Norit1600 catalyst (approaching 80%, for reaction temperature 251 °C, Tables S1–S4) and rather smooth changes in the selectivity with adding platinum. In addition to the regular increase in $CH_4$ selectivity, the Pt-richer catalysts demonstrated the formation of fluoromethane and even fluoroform. Smooth variations of product selectivity with Pd-Pt alloy composition would indicate a lack of synergy in the hydrodechlorination behavior of the bimetallic system, but rather a linear cumulative action of these alloys.

These relations resemble very much the previous selectivity patterns obtained for the same catalysts precalcined at 300 °C for 1 and 2 h [17].

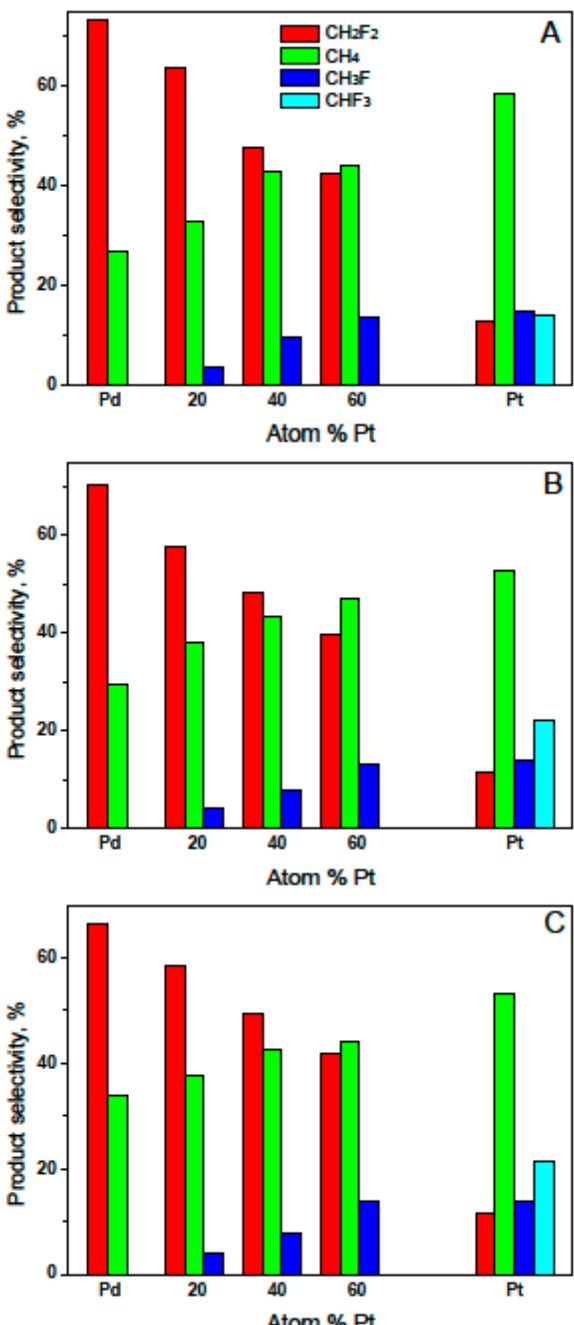

**Figure 1.** CHClF$_2$ hydrodechlorination over 2 wt % Pd–Pt(acac)/Norit1600 catalysts at 272 °C. The effect of nominal bimetal composition on product selectivity. (**A**)—after precalcination at 320 °C (1 h), (**B**)—after precalcination at 350 °C (1 h), and (**C**)—after precalcination at 400 °C (0.25 h).

Still more interesting results relate to the evolution of catalytic activity of Pd-Pt/Norit1600 systems produced by different precalcination conditions. Figure 2 shows that the increase in precalcination temperature produced an increase in the activity of all catalysts, however, the Pd-rich catalysts (2 wt % Pd*100*(acac)/Norit1600, 2 wt % Pd*80*Pt*20*(acac)/Norit1600, and 2 wt % Pd*100*(Cl)/Norit1600) gained the most. The turnover frequency of 2 wt % Pd*100*(acac)/Norit1600 experienced a spectacular increase, by nearly an order of magnitude, from 0.00041 to 0.00263 s$^{-1}$.

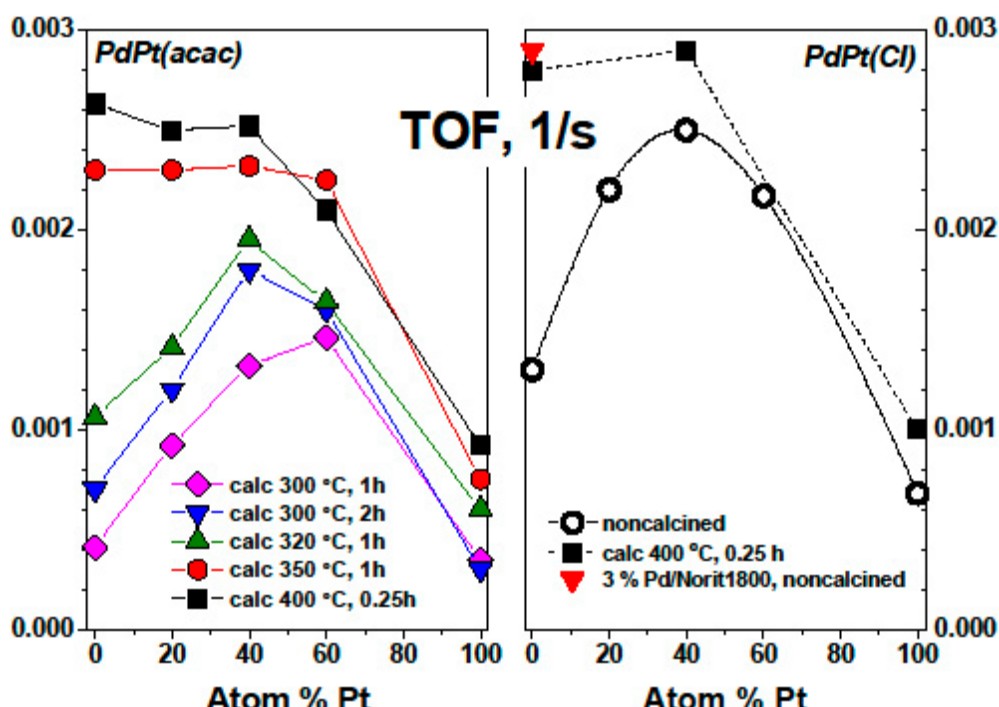

**Figure 2.** The effect of precalcination temperature on the relation between the turnover frequency (TOF) of Pd-Pt/Norit1600 catalysts and Pd-Pt alloy composition. The results for Pd-Pt(acac) catalysts precalcined at 300 °C and for Pd-Pt(Cl) non-calcined samples were taken from [17].

Our earlier study [17] revealed that the Pd-Pt catalysts prepared from metal acetylacetonates were less active than the ex-chlorides ones. This is recalled in Figure 2. The reasons for this difference could not be seen explained by contamination with carbon containing species because the ex-acac catalysts were subjected to careful oxidation at 300 °C, whereas the ex-chloride samples were not preoxidized. Nevertheless, precalcination of the ex-chloride samples at 400 °C also increased their reactivity (Figure 2). However, a strong maximum for 2 wt % Pd*60*Pt*40*(Cl)/Norit1600 for non-calcined ex-Cl samples was flattened upon precalcination. The red triangle in Figure 2 represents the catalytic behavior of 3 wt % Pd/Norit1800, the catalyst prepared using the Norit carbon preheated in helium at 1800 °C [23]. Its basic characteristics is in the Supplementary Materials (Table S4 and SET S1). Its good catalytic performance in $CHClF_2$ hydrodechlorination was achieved without calcination in oxygen. On the other hand, additional calcination practically did not change its catalytic activity (result not shown).

The TOF values were calculated taking into account the metal dispersions determined by hydrogen chemisorption, H/(Pd + Pt). Metal dispersion did not change significantly with the change in precalcination conditions (data in Tables S1–S4 vs. relevant dispersion values reported in [17]). In line with the chemisorption studies, X-ray diffraction (XRD) examination of reduced Pd/Norit1600 catalysts confirmed the presence of only minor changes in the metal crystallite size at different precalcination temperatures (Figure 3).

Thus, if precalcination of 2 wt % Pd*100*(acac)/Norit1600 does not lead to a significant change in metal dispersion, the large increase in the activity of this catalyst could result from additional cleaning of its surface from carbon. The problem of decorating the surface of palladium with carbon from the carrier has been known for years [20–22]. The relative consistency of the size of the palladium particles resulting from the metal dispersion measurement ($d_{Pd}$ = 1.12/(H/Pd), [24]) and the size of the crystallites Pd (Figure 3) suggests that metal surface decoration does not take place here. It should be added that the pretreatment of all ex-acac catalysts included their initial precalcination at temperatures $\geq$300 °C, a step suggested for carbon elimination from the palladium surface [20–22]. Nevertheless,

we decided to carry out TPO measurements of Norit-supported palladium catalysts that could shed light on the forms of carbon eliminated in the course of oxidation.

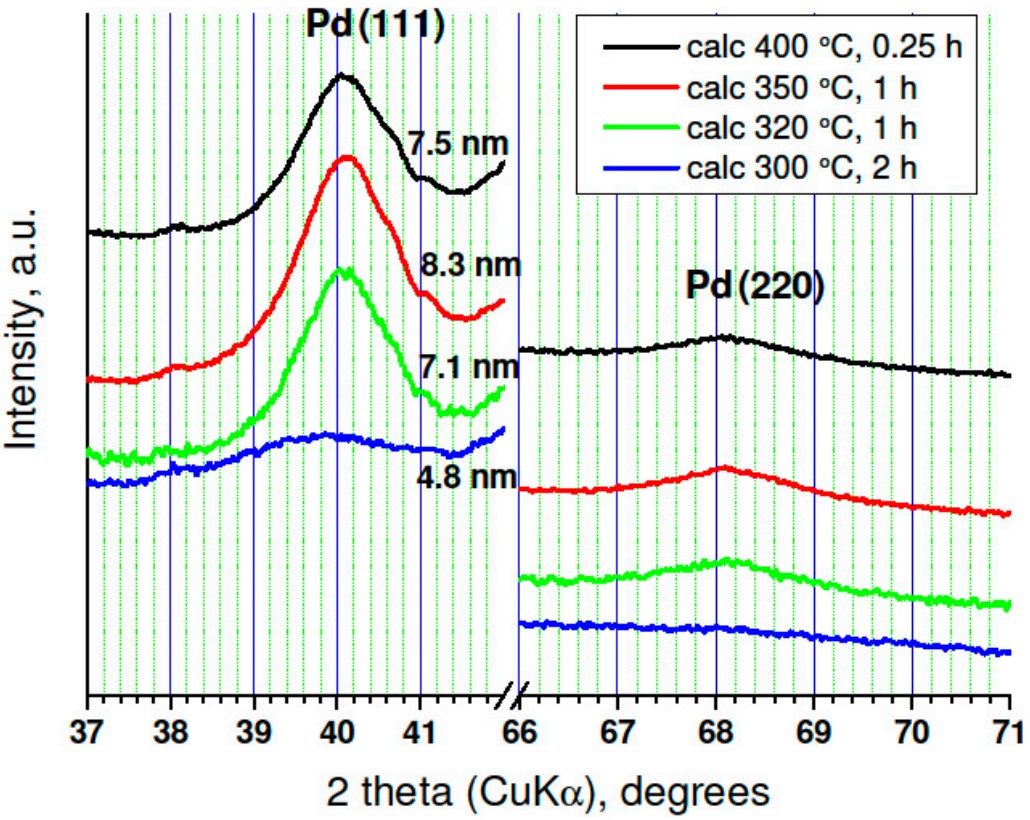

**Figure 3.** XRD profiles of 2 wt % Pd/Norit1600 catalysts precalcined at different temperatures and reduced at 400 °C for 1 h. Pd crystallite size was determined from the (111) XRD line broadening using the Scherrer formula.

The results of TGA-TPO studies of 2 wt % Pd*100*(acac)/Norit1600, 2 wt % Pt*100*(acac)/Norit1600, and Norit1600 carbon are shown in Figure 4. As carbon oxidation (seen as evolution of $CO_2$) was catalyzed by Pd and Pt, a considerable decrease in the burn-off temperature of carbon, by 200 °C, compared to the oxidation of bare Norit1600 carbon can be observed. However, the shape of TPO profiles for 2 wt % Pd*100*(acac)/Norit1600 and 2 wt % Pt*100*(acac)/Norit1600 did not contain any additional characteristic peaks, which were observed in TPO of Pd/C catalysts by Tengco et al. [21] and could be ascribed to the burn-off of carbon species released from the surface and subsurface layer of palladium. However, our observations were consistent with the results of the cited work because the preoxidation of 2 wt % Pd*100*(acac)/Norit1600 at 300 °C used in our preliminary catalyst pretreatment appears to be sufficient in order to remove superficial carbon from palladium. Therefore, massive carbon removal (TG) accompanied by vast $CO_2$ evolution (*m/z* 44) at temperatures above 300 °C (especially at 400 °C, see dotted line in Figure 4) should be attributed to the oxidation of support carbon in close contact with metal, called 'proximate' carbon in [21].

There are some differences in the course of TPO profiles for both metals. The onset of this increase is for the Pt/C catalyst delayed by 30 °C compared to the behavior of Pd/C (Figure 4). This 'delay' would be due to the fact that there is less 'proximate' carbon near platinum, which, in turn, may be due to the lower adherence of platinum to carbon. It is known that carbon nanotubes are better wetted by palladium than by platinum [25]. However, after exceeding the temperature of 400 °C, platinum shows a higher rate of carbon oxidation. In conclusion, we suggest that the massive firing of pore walls catalyzed

by the presence of entrapped metal particles removes steric obstacles for free access of gas reactants to the entire metal surface, making it more active.

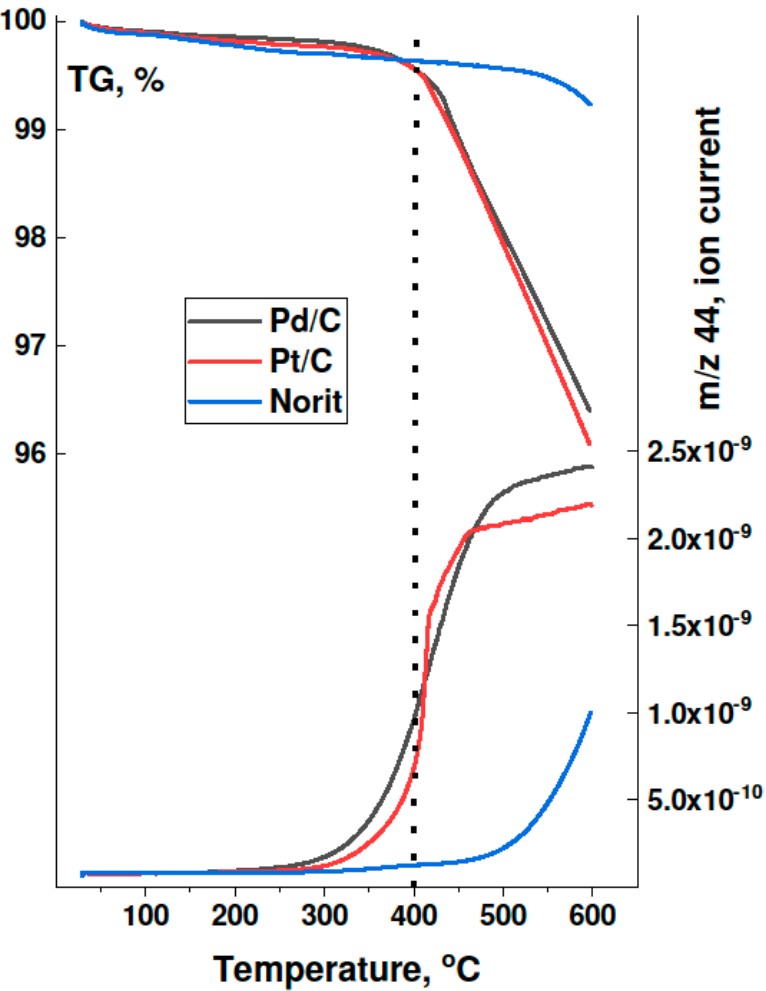

**Figure 4.** TG-TPO profiles (left axis) of 2 wt % Pd*100*(acac)/Norit1600 (black), 2 wt % Pt*100*(acac)/Norit1600 (red), and Norit 1600 carbon (blue). Evolution of $CO_2$ (right axis) monitored during TGA-TPO at a 5 °C/min ramp.

Considerable removal of the 'proximate' carbon at 350–400 °C should generate changes in the pore structure of the materials and we decided to check it. The results from the studies of physical adsorption of nitrogen at liquid nitrogen temperatures are presented in Figures 5 and 6, and in Table 1. Figure 5 shows $N_2$ adsorption isotherms for 2 wt % Pd*100*(acac)/Norit1600, 2 wt % Pt*100*(acac)/Norit1600, and for unloaded Norit1600. All nitrogen adsorption/desorption isotherms show the hysteresis loop of H3 type, indicating the presence of slit-shaped pores. The Figures show trends associated with loading carbon with the metals (impregnation, red lines) and carbon removal by burning (black lines). A relative departure of the red isotherm (for the catalyst) from the blue one (for Norit) is larger for Pd/C than for Pt/C. The results in Table 1 show that the impregnation of Norit1600 with platinum only slightly decreased the BET surface area and pore volume of the system. In the case of palladium, the corresponding changes were larger. This may result from the fact that, on average, the 2 wt % Pt*100*(acac)/Norit1600 catalyst contained smaller metal particles than the 2 wt % Pd*100*(acac)/Norit1600 (3 vs. 7 nm, Table 1 in [17]). However, it should be noted that the volume of introduced platinum was 1.5 times smaller than the volume of introduced palladium (with the same 2 wt % metal loading in all catalysts).

**Table 1.** Surface areas and pore volume data from $N_2$ adsorption isotherms.

| Property | | Norit1600 | 2 wt % Pd*100*(acac)/Norit1600 | | 2 wt % Pt*100*(acac)/Norit1600 | |
|---|---|---|---|---|---|---|
| | | | Noncalcined [a] | Calcined [b] | Noncalcined [a] | Calcined [b] |
| BET surface area, $m^2/g$ | | 167.8 | 137.3 | 192.6 | 164.1 | 219.7 |
| BJH pore volume, $cm^3/g$ | from adsorption | 0.254 | 0.229 | 0.292 | 0.242 | 0.286 |
| | from desorption | 0.259 | 0.247 | 0.302 | 0.250 | 0.294 |
| $t_{plot}$ micropore volume, $cm^3/g$ | | 0.00577 | 0.00188 | 0.00335 | 0.00397 | 0.00667 |

[a] After preparation and initial pretreatment (precalcined at 300 °C, 1 h, not postcalcined at 400 °C); [b] After additional calcination at 400 °C for 0.25 h.

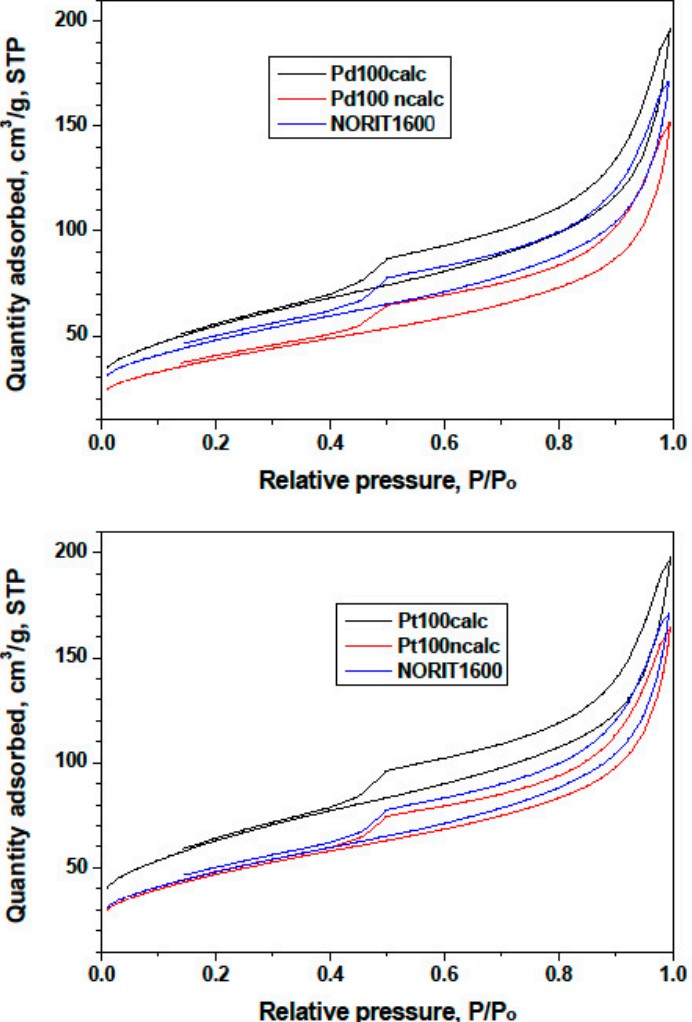

**Figure 5.** Physical adsorption–desorption isotherms of $N_2$ on Pd/Norit1600 (**top**) and Pt/Norit1600 (**bottom**). The isotherm for Norit1600 is included in the graphs.

After oxygen treatment at 400 °C, the $N_2$ adsorption isotherm for Pt/C is more markedly shifted toward higher $N_2$ uptake than the isotherm for 2 wt % Pd*100*(acac)/Norit1600. This may result from the higher catalytic activity of platinum (than palladium) in carbon oxidation. Platinum appeared to be a uniquely good metal in soot oxidation [26]. Overall, interesting conclusions follow from the evolution of micropore volume ($t_{plot}$). Impregnation of Pd leads to a loss of micropore volume by 0.00389 $cm^3/g$ (=0.00577–0.00188, Table 1). Although the total volume of introduced palladium (0.0017 $cm^3/g$) would be accommodated in the micropores, it is certain that only a part of Pd nanoparticles blocks the

micropores, probably at their outlets. The analysis of the distribution of metal particle size (Table 1 in [17]) indicated the occurrence of a large range of differently sized metal particles, some of them would not fit the micropores. Pd/C precalcination at 400 °C for 15 min recovers a substantial part of micropores, suggesting that very small Pd nanoparticles would now be more accessible to the gas reactants.

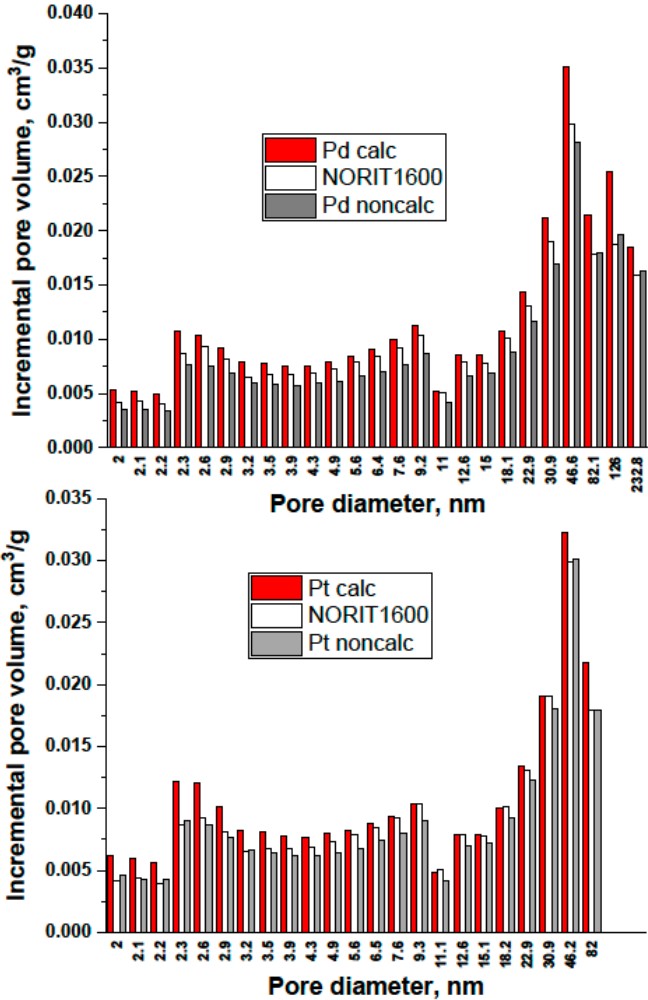

**Figure 6.** Pore size distribution of differently pretreated 2 wt % Pd*100*(acac)/Norit1600 (**top**) and 2 wt % Pt*100*(acac)/Norit1600 (**bottom**) catalysts.

Figure 6 presents variations in the pore size distribution in impregnated and calcined Norit1600, 2 wt % Pd*100*(acac)/Norit1600, and 2 wt % Pt*100*(acac)/Norit1600 catalysts. Again, the pore structure of the 2 wt % Pt*100*(acac)/Norit1600 catalyst did not differ much from the structure of Norit1600 (black vs. white bars in Figure 6). For 2 wt % Pd*100*(acac)/Norit1600, the differences were larger: all black bars (with exception for very large pores) were smaller than the white ones. This would indicate a widespread distribution of Pd particles over Norit1600, i.e., the metal location throughout the entire pore structure. On the other hand, the calcination of these catalysts led to a roughly uniform volume increase in all pores, regardless of their size. Earlier studies [27] on graphite oxidation by Pd and Pt showed that all active metal particles, irrespective of their size, gasify the same amount of carbon per unit time under given experimental conditions. In the case of 2 wt % Pt*100*(acac)/Norit1600, a somewhat higher increase in the volume of small pores with 2.2–2.8 nm diameter can be observed. This pore size corresponded with the mean size of metal particles in the reduced 2 wt % Pt*100*(acac)/Norit1600) catalyst.

From the results of the physical characterization of 2 wt % Pd*100*(acac)/Norit1600 and 2 wt % Pt*100*(acac)/Norit1600 catalysts, one could conclude:

(1)  Oxidation of Pd-on-highly preheated Norit catalyst, which led to nearly an order of magnitude increase in catalytic activity, could not be the result of marked changes in metal dispersion changes because such changes were not found.

(2)  Possible decontamination of palladium surface from carbon by oxidation at 350–400 °C was also rejected as a basic reason for the activity increase. Preliminary precalcination of 2 wt % Pd*100*(acac)/Norit1600 samples at 300 °C for 1–2 h should remove the carbon from the metal surface [21,22]. TPO profiles of such catalysts did not contain any signs of presence of such carbon species.

(3)  TGA-MS studies show the beginning of a massive removal of a "proximate" carbon at the temperature 350 °C. Metal-catalyzed burning of carbon support changes the pore structure of Norit1600. In particular, the micropore volume was vastly increased along with catalyst oxidation. Metal nanoparticles, wetting small pores of the support, presumably lose contact with the pore walls as a result of oxidation. Such a catalyst represents enhanced reactivity, proving the accessibility of the active sites to reactants.

Figure 2 showed the effect of calcination temperature on the catalytic performance in CHClF$_2$ hydrodechlorination. By increasing the activity of Pd and Pd-rich alloys as an effect of an increased temperature of calcination, the relation between TOF and PdPt alloy composition changes its primary course. The Pd-Pt/C samples precalcined at 300 and 320 °C show the sharp maximum at 40 at.% Pt, but after precalcination at ≥350 °C this maximum disappeared. This means that the synergistic effect in catalytic action of Pd and Pt reported in our previous paper [17] ceases to exist. The previously discussed product selectivity data were also in opposition to the synergy. Figure 7 summarizes the results of our previous and present study. The course of the relation between the TOF and Pd-Pt alloy concentration after burning carbon from pore walls did not contain any maxima but tended to reflect the inferred surface concentration of Pd in Pd-Pt alloys (inset in Figure 7 based on [28]). This suggests on a statistical basis that the course of CHClF$_2$ hydrodechlorination on Pd-Pt surfaces would be controlled by the presence of Pd on the surface, with active sites composed of single Pd atoms and not, for example, on Pd-Pd doublets. Single Pd atoms were found to be highly active sites for the hydrodechlorination of 4-chlorophenol [29].

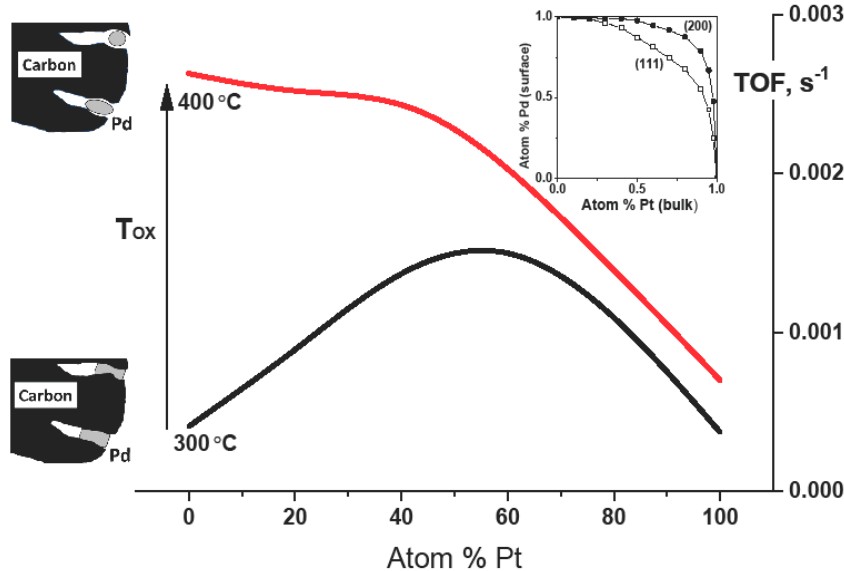

**Figure 7.** Changes in the catalytic behavior of Pd-Pt/Norit1600 catalysts in chlorodifluoromethane hydrodechlorination caused by catalyst precalcination. Inset: the relation between surface and bulk composition of Pd-Pt alloys adapted from Rousset et al. [28].

Good catalytic performance of non-calcined 3 wt % Pd/Norit1800 catalyst used in our earlier work [23], shown in Figure 2 (red triangle), should be commented on. A relatively

low BET surface area and pore size distribution in this sample (SET S1 in Supplementary Materials) appeared to favor good access of reactants to active sites. The very small micropore volume in this impregnated catalyst (0.0010 cm$^3$/g, SET S1), in combination with the lack of micropores in carbons heated at 1800 °C [30], confirmed the absence of Pd particles in micropores.

We decided to investigate the 3 wt % Pd/Norit1800 catalyst in CHClF$_2$ hydrodechlorination using somewhat intensified reaction conditions, i.e., relatively low GHSV values. Figure 8 shows good stability and very good selectivity to CH$_2$F$_2$ (90–95%). This result calls for further studies with use of very highly pretreated carbons.

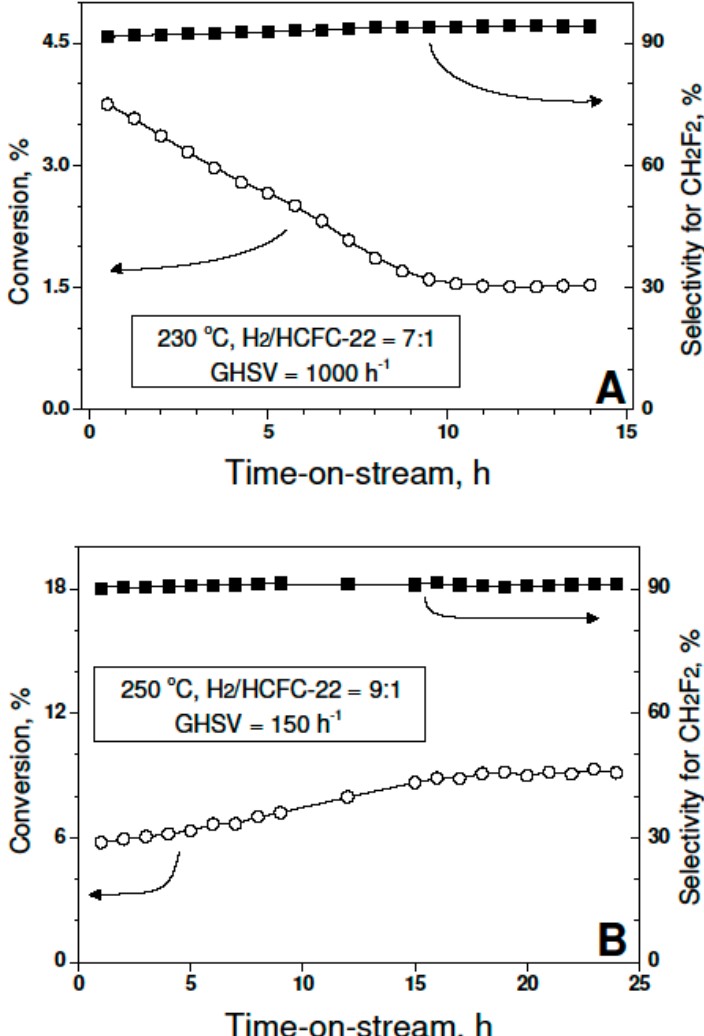

**Figure 8.** Time-on-stream behavior (conversion of CHClF$_2$ and selectivity to CH$_2$F$_2$) of 3 wt % Pd/carbon at different reaction conditions incorporated in (**A**,**B**) sections. Norit RO 08 carbon support was heated in helium at 1800 °C for 2 h. Reaction conditions (temperature, H$_2$/HCFC-12 ratio and GHSV) are included in figure frames.

## 3. Methods

### 3.1. Catalyst Preparation

The preparation and basic pretreatment were described in our previous paper [17]. Briefly, Norit RO 0.8 activated carbon (CAS # 7440-44-0), pretreated in He at 1600 °C for 2 h (referred in the text as Norit1600), was coimpregnated with a toluene (CAS 108-88-3, analytical purity from Chempur, Piekary Śląskie, Poland) solution of the metal acetylacetonates (Pd(acac)$_2$, CAS 14024-61-4 from Sigma-Aldrich, St. Louis, MO, USA, and Pt(acac)$_2$, CAS 15170-57-7 from abcr GmbH, Karlsruhe, Germany, both 99% pure) to prepare monometallic

Pd and Pt and three bimetallic Pd-Pt catalysts (atomic ratio of Pd-to-Pt: 80:20, 60:40, and 40:60), designed as Pd*X*Pt*Y*, where *X* and *Y* stand for atomic percentages of Pd and Pt in the metal phase. The overall metal loading was 2 wt %.

As mentioned in the previous report [17], the removal of the organic part of acetylacetonates achieved by a 1 h calcination in the flow of 1% $O_2$/Ar (50 cm$^3$/min, at STP) at 300 °C. In our previous work, the precalcined catalysts were subjected to reduction in flowing $H_2$/Ar mixture (20 cm$^3$/min, STP) at 400 °C before the reaction. In the present work, the stored catalysts were subjected to more severe secondary calcinations in $O_2$/Ar at 320 °C, 350 °C, and 400 °C.

Some kinetic runs were performed with a few Pd, Pt, and Pd-Pt catalysts prepared from the chloride precursors. Their preparation was also described in [17].

### 3.2. Catalytic Tests

The hydrodechlorination of chlorodifluoromethane was performed in a flow system equipped with a gradientless glass reactor, as described previously [17]. Before each reaction run, the catalyst (0.20 g sample) was exposed to flowing 1% $O_2$/Ar (20 cm$^3$/min, STP) at 320 °C and 350 °C for 1 h, or at 400 °C for 15 min. The precalcination was followed by a short purge with Ar at 400 °C, and the reduction in 10% $H_2$/Ar (20 cm$^3$/min, STP) at 400 °C for 1 h. After reduction, the catalysts were cooled in $H_2$/Ar flow to the desired initial reaction temperature, i.e., 270 °C. For a typical reaction run, the total flow of the reactant mixture was 48 cm$^3$/min and consisted of CHClF$_2$ (1 cm$^3$/min), H$_2$ (8 cm$^3$/min), and Ar (39 cm$^3$/min), fixing the GHSV at 5760 h$^{-1}$. This high value allowed the maintenance of low conversions, usually <3%, and minimized secondary reactions. The reaction was carried out until a steady state was achieved at 270 °C (16 h). Then, the reaction temperature was gradually decreased to 260 °C and 250 °C and new experimental points were collected. A typical run lasted 20 h. The post-reaction gas was analyzed by gas chromatography [17]. Product selectivities were defined as (C$_i$/ΣC$_i$) × 100%, where C$_i$ is the molar concentration of the detected product i. Some kinetic runs were performed with a few precalcined Pd-Pt/C catalysts prepared from metal chlorides. In those cases, the reduction in $H_2$/Ar at 400 °C lasted 3 h. Finally, the 3 wt % Pd supported on Norit carbon preheated at 1800 °C, used in our previous studies [23], was also tested.

### 3.3. Catalyst Characterization by Hydrogen Chemisorption, XRD, TPO, and Physical Adsorption (BET, Pore Structure)

As in our previous work [17], metal (Pd, Pt) dispersion was determined using hydrogen chemisorption, following the procedure described in [31]. A reduced and outgassed catalyst was flushed with H$_2$ at 70 °C, and the amount of adsorbed hydrogen was measured by desorption in argon, using a 20 °C/min temperature ramp.

XRD studies of prereduced and, in a few cases, post-reaction samples of Pd-Pt/C catalysts were conducted on a Rigaku–Denki diffractometer using Cu*Kα* radiation. The samples were tested using a step-by-step technique, at 2 theta intervals of 0.05° and a counting time of 10 s for each point.

TGA analysis combined with mass spectrometry was performed using the STA449C thermobalance (NETZSCH, Selb, Germany) with the QMS 403C Aeolos quadrupole mass spectrometer. Measurements were carried out on the samples of Norit1600, Pd*100*(acac)/C, and Pt*100*(acac)/C of 35 mg each, which were heated at a 5 °C/min ramp from 30 to 600 °C, under a 1% $O_2$/Ar mixture flow (100 cm$^3$/min, at STP). The temperature, mass changes and $m/z$ signals of relevant gases ($O_2$, CO, $CO_2$, and $H_2O$) were continuously recorded during these TPO measurements.

Surface areas and porosities were measured with an ASAP 2020 instrument from Micromeritics, employing the BET (Brunauer–Emmett–Teller), t-Plot, BJH (Barrett–Joyner–Halenda), and HK (Horvath–Kawazoe) methods and using nitrogen as the adsorbate. Before measuring the adsorption isotherm at −196 °C, the sample was kept at 200 °C for 5 h in a vacuum to clean its surface.

## 4. Conclusions

Pd-Pt catalysts of 2 wt % supported on Norit activated carbon preheated at 1600 °C were reinvestigated in the hydrodechlorination of chlorodifluoromethane. An additionally adopted catalyst oxidation pretreatment, at 320, 350, and 400 °C, caused an order of magnitude increase in TOF for the monometallic Pd/C, from $4.1 \times 10^{-4}$ to $2.63 \times 10^{-3} \, \mathrm{s}^{-1}$. Neither possible changes in metal dispersion nor metal decontamination from carbon were found responsible for this effect. TGA-MS studies show the beginning of massive removal of a "proximate" carbon at the temperature 350 °C, i.e., at the onset of the increase in catalytic activity. Metal-catalyzed burning of carbon support modifies the pore structure of Norit1600. The micropore volume in particular is vastly increased as an effect of catalyst oxidation. Metal nanoparticles, wetting the small pores of the support, presumably lose contact with the pore walls as a result of oxidation. Such a catalyst demonstrates enhanced reactivity, hence proving good access for reactants to the catalytically active sites. Therefore, the reactivity tuning of Pd/C catalysts using oxidation is not due to changes in metal dispersion but from unlocking the active metal surface, originally inaccessible in Pd particles tightly packed in the pores of carbon. In agreement with our speculations, a non-calcined Pd/C catalyst supported on the carbon preheated at 1800 °C showed good catalytic behavior. Separate runs with 3 wt % Pd/carbon under somewhat intensified reaction conditions presented very good catalyst stability and excellent selectivity to $CH_2F_2$ (>90%).

In our discussion we did not consider the possible inhibiting role of residual chloride originating from the metal precursor for two reasons. First, the ex-chloride catalysts were generally more active than the ex-acac ones. Second, during the reaction of $CHClF_2$ hydrodechlorination, the metal surface, regardless of its origin, is covered by a variety of active Cl/F and HCl/HF species, so the possible role of residual chloride from the precursor would be difficult to distinguish.

**Supplementary Materials:** The following are available online at https://www.mdpi.com/article/10.3390/catal11050525/s1, Table S1: $CHClF_2$ hydrodechlorination on Pd-Pt/(acac)/Norit1600 catalysts precalcined at 320 °C for 1 h. TOFs, selectivities and apparent energies of activation, Table S2: $CHClF_2$ hydrodechlorination on Pd-Pt/(acac)/Norit1600 catalysts precalcined at 350 °C for 1 h. TOFs, selectivities and apparent energies of activation, Table S3: $CHClF_2$ hydrodechlorination on Pd-Pt/(acac) (/Norit1600 catalysts precalcined at 400 °C for 15 min. TOFs, selectivities and apparent energies of activation, Table S4: $CHClF_2$ hydrodechlorination on Pd-Pt/Norit catalysts prepared from palladium and platinum chlorides. TOFs, selectivities and apparent energies of activation, SET S1: Characteristics of 3 wt % Pd/Norit1800 catalyst; Figure S1: XRD profiles from Norit1800 and 3 wt.% Pd/Norit1800 catalysts; Figure S2: $N_2$ adsorption-desorption isotherms of $N_2$ on Norit1800 carbon (top) and pore size distribution in Norit1800 activated carbon (bottom).

**Author Contributions:** M.R. was responsible for catalysts preparation and characterization, kinetic studies, experiment planning and manuscript writing; W.J. characterized the catalysts by XRD; W.R.-P. and M.Z. were responsible for preparation and characterization of activated carbon, BET and TGA-MS studies; Z.K. was responsible for conceptual work, experiment planning and overall care about manuscript writing. All authors have read and agreed to the published version of the manuscript.

**Funding:** Research Project # 2016/21/B/ST4/03686 from the National Science Centre (NCN), Poland.

**Data Availability Statement:** Data is contained within the article or Supplementary Material.

**Conflicts of Interest:** The authors declare no conflict of interest.

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
