# Peer review of "Chlorodifluoromethane Hydrodechlorination on Carbon-Supported Pd-Pt Catalysts. Beneficial Effect of Catalyst Oxidation"

_catalysts, doi:10.3390/catal11050525_

Round 1

Reviewer 1 Report

The manuscript deals with the Pd-Pt catalysts for hydrodechlorination of CHClF2. The results are clearly stated in the manuscript. There are minor remarks:

COMMENTS

1). A period (full stop) is not put after manuscript title.

2). The elemental composition (Pd, Pt) of each catalysts should be determined.

3). Page 2 line 76: «2. Results» should be replaced by «2. Results and Discussion».

4). Caption of Figure 1: What type of support was used to prepare the catalysts?

5). Page 4 line 131: How was the metal crystallite size determined? Or the size of the metal particles???

6). Is carbon dioxide the only product of support oxidation? What gases were still emitted?

7). The reagent adsorption depends on the nature and amount of functional groups on the support surface. Has the composition of functional groups changed during heat treatment? Can this factor affect the reaction rate?

8). Page 10 lines 251-253: The authors argue that «This suggests that the course of CHClF2 hydrodechlorination on Pd-Pt surfaces would be controlled by the presence of Pd on the surface, with active sites composed of single Pd atoms.». However, the evidence is lacking. References to articles should be added.

9). «3. Methods»: The «Materials» section should be added. CAS or main component content should be given for each of the compounds used in the experiments.

10). How was selectivity for each catalyst detected?

11). Could acetylacetonate be the reason for the carbonization of the active component of the catalyst at 300 ° C? This factor should be discussed specifically.

I recommend this paper to be accepted for the publication with minor revision.

Best regards, Reviewer.

Reviewer 2 Report

The paper is a continuation of the author’s previous work on a highly important topic. It is somewhat long as compared to its content therefore the main message is not very clear. The following comments should be considered before the publication:

1) The abstract needs to be made more informative. Mention metal loading, dispersion and other relevant catalyst parameters. Mention how the catalysts were characterised. A sentence “the performance of the rest of the Pd-Pt/C catalysts changes less…  ” should be rewritten. “Previous work” in the abstract should be cited explicitly. Finally mention reaction conditions and typical TOF and selectivity values

2) The catalyst codes should be made consistent throughout the manuscript. There are 3 different codes are currently used: Pt100(acac), 2 wt.% Pt/Norit1600 and Pd/C for the same material. This confusing and difficult to follow.

3) The percentage should be explained whether this is wt% or vol% or atom%.

4) Figure 6. Why the x-axis has a nonlinear scale? This plot can be moved into supplementary information.

5) A possible reason why it takes 20 hours to reach a steady-state should be explained in more details.

6) Could you also add metal particle size distribution plots before and after oxidative treatment?  

Experimental section.

6) Line 303. “The post-reaction gas was analyzed by gas chromatography”. The analysis should be described in more details or a reference to a previous paper should be provided.

7) Line 306. “the 3 wt.% Pd supported on Norit carbon”. Please specify whether this is nominal or actual metal loading. Nothing is mentioned how the metal loading was measured in the samples.

Reviewer 3 Report

At the present work, authors have studied the effect of pre-calcination step in the activity and selectivity of Pd-Pt catalysts supported on carbon (preheated to 1600 °C) in the hydrodechlorination reaction of CHClF2. In the study, a significant improvement of the TOF of the catalysts is obtained increasing the temperature of the pre-calcination step, whereas the selectivity does not change in a great extent. The cause of this improvement in the activity is studied by the authors through different technics analyzing, among others: the metal dispersion, the crystalline structure and the pore size and distribution of the catalysts; the loss of carbon during the pre-calcination step. Basing on these experiments, authors have hypothesized that the carbon elimination from the pore walls catalyzed by the metal during the pre-calcination step let the access of the reactants to higher number of active sites.

Comments:

-Since the point of view of novelty, these kind of catalysts have been previously developed by the authors and used in the same reaction in a previous work. Moreover, the beneficial effect of the pre-calcination step has also been observed (in a very minor extent) in this previous work (Figure 2 in the manuscript, PdPt(acac), 300 °C and 1 h vs 300 °C and 2 h of pre-calcination). In the present work, however, the effect of pre-calcination has been studied more in detail by the authors, carrying out this step at higher temperatures.

-The extensive study of the pre-calcination step in this work has provided a significant increment of the turnover frequency of the catalysts, that presents great interest.

-Basing on the experiments developed, authors have proposed that the improvement in the activity of the catalysts is related to changes happened in the pores, and not due to changes in the metal dispersion or in the contamination of the metal surface with superficial carbon. This presents great interest.

Specific comments:

-The reference 1 corresponds to an entry in “Wikipedia” and should be changed by a more rigorous information source.
